# Deep-Learning Based Virtual Stain Multiplexing Immunohistochemistry Slides – a Pilot Study

**Oded Ben-David**                                    ODED.BEN-DAVID@AGILENT.COM
**Elad Arbel**
**Daniela Rabkin**
**Itay Remer**
**Amir Ben-Dor**
*Agilent Research Laboratories, Tel Aviv, Israel*

**Sarit Aviel-Ronen**
*Adelson School of Medicine, Ariel University, Ariel, Israel*

**Frederik Aidt**
**Tine Hagedorn-Olsen**
**Lars Jacobsen**
*Agilent Technologies, Glostrup, Denmark*

**Kristopher Kersch**
*Agilent Technologies, Carpinteria, US*

**Anya Tsalenko**
*Agilent Research Laboratories, Santa Clara, US*

## Abstract

In this paper, we introduce a novel deep-learning based method for virtual stain multiplexing of immunohistochemistry (IHC) stains. Traditional IHC techniques generally involve a single stain that highlights a single target protein, but this can be enriched with stain multiplexing. Our proposed method leverages sequential staining to train a model to virtually stain multiplex additional IHC on top of a digitally scanned whole slide image (WSI), without requiring a complex setup or any additional tissue sections and stains. To this end, we designed a novel model architecture, guided by the physical sequential staining process which provides superior performance. The model was optimized using a custom loss function that combines mean squared error (MSE) with semantic information, allowing the model to focus on learning the relevant differences between the input and ground truth. As an example application, we consider the problem of detecting macro-phages on PD-L1 IHC 22C3 pharmDx NSCLC WSIs. We demonstrated virtual stain multiplexing CD68 on top of PD-L1 22C3 pharmDx stained slides, which helps to detect macrophages and distinguish them from PD-L1+ tumor cells, which are often visually similar. Our pilot-study results showed significant improvement in a pathologist's ability to distinguish macrophages when using the virtually stain multiplexed CD68 decision supporting layer.

**Keywords:** Immunohistochemistry, Virtual Stain, Multiplexing, Deep Learning, Macrophages, NSCLC.

## 1 Introduction

"Tissue is the issue" is a theme that serves as a guideline in nowadays pathology practice, that requires ever-growing information deduced from tissue samples, while their size keeps

getting smaller. Extracting maximum information from tissue samples is a common challenge in pathology in the era of personalized medicine that necessitates the use of increasing number of immunohistochemical (IHC) stains. Combined with the increasing use of small biopsies as the main tissue material, this puts a significant limit on the number of different IHC stains which can be run. While this challenge can be addressed in research using fluorescent IHC multiplexing or mass spectrometry IHC, in routine diagnostic pathology the common practice is to stain multiple consecutive sections with different immunostains at the cost of labor, reagents and tissue.

Digital pathology offers an array of alternative solutions as it enables to digitally unmix (Ruifrok and Johnston, 2001) and superimpose staining from one section to another, to create virtual multiplexed layers (Huss and Grunkin, 2022; Visiopharm, 2023) or otherwise to project AI inferred specific cell density maps (Bloom et al., 2022). Virtual multiplexing methods, however, suffer from the same two main draw-backs as traditional pathology. First, the method requires additional tissue sections and reagents, adding to cost and complexity. Second, the consecutive tissue sections are at least 3-5 microns apart resulting in an increasingly growing distance between slides with each additional stained section. Thus, sections do not contain the exact same cells.

With recent advances in deep learning based models, a different approach called virtual staining has emerged (Owkin, 2020; de Haan et al., 2021). Virtual staining methods use generative adversarial networks (GANs) to unlock information from existing Hematoxylin-Eosin (HE) stained tissue patterns without the need for physically staining additional tissues. Pushing this approach even further enables the inference of immunohistochemical staining patterns using unlabeled tissue (Rivenson et al., 2019; Zhang et al., 2020; Pradhan et al., 2021). For example, Bai et al. (2022) developed a method that generates a virtual HER2 IHC whole slide image (WSI) by recording several auto-fluorescence images of an unlabeled tissue section.

A prime example of the benefits of combining information from multiple stains is the detection of macrophages in PD-L1 22C3 pharmDx stained non-small cell lung cancer (NSCLC) tissues.

PD-L1 IHC 22C3 pharmDx (GE006) is an FDA approved qualitative immunohistochemical assay intended for use in the detection of PD-L1 protein in formalin-fixed, paraffin-embedded (FFPE) non-small cell lung carcinoma NSCLC tissue. PD-L1 protein expression in NSCLC is determined by using Tumor Proportion Score (TPS) (Herbst et al., 2016), which is the percentage of viable tumor cells showing partial or complete membrane staining at any intensity. The TPS scoring protocol calls for the exclusion of the staining of immune cells including macrophages (Technologies, 2021). Since macrophages may show morphological similarity to NSCLC cells, differentiating positive PD-L1 staining of macrophage from that of tumor cell is often challenging (Paces et al., 2022). Although some morphological features distinguish macrophages from tumor cells, combined staining of PD-L1 IHC 22C3 pharmDx with a separate IHC stain such as CD68 can highlight the macrophages and assist clinical pathologists in scoring more accurately.

Beck et al. (2019) developed an automatic PD-L1+ detection method for urothelial carcinoma, achieving strong correlation (0.837) with pathologist consensus scores on tumor cells. Although not required for clinical purposes, they found that the detection of

macrophages is still a challenging task, with inter-observer agreement/correlation as low as 0.287.

In this paper, we present a novel method called virtual stain multiplexing, which combines virtual staining and virtual multiplexing. We apply this method to the problem of detecting macrophages on PD-L1 IHC 22C3 pharmDx NSCLC WSIs.

The novel architecture of our proposed method is guided by the physical sequential staining process and is trained using a loss function that combines mean squared error (MSE) with semantic information. Our approach of incorporating semantic information into the training process proved to be a crucial component in achieving the desired performance.

We trained an AI-based model using sequentially stained NSCLC WSIs to infer virtual CD68 IHC from an input stained with PD-L1 IHC 22C3 pharmDx. The inferred virtual stain is then combined with the input to yield the virtual stain multiplexed output. As a pilot-study, we tested the effectiveness of our model by comparing the performance of a pathologist in detecting macrophages in PD-L1 IHC 22C3 pharmDx stained NSCLC with and without using the CD68 virtual stain as a decision-supporting layer. We found that the addition of the virtual stain significantly improved the performance of the pathologist in detecting macrophages.

## 2 Methods

Sequential staining is a method for adding an additional immunohistochemical (IHC) stain (stain 2), to tissues which were already stained (stain 1) Moreover, according to the Beer-Lambert law for absorption of light passing through a medium [15], the optical density of stained tissue is linearly dependent on the local concentration and absorption coefficient spectrum of chromogen.

We designed a model architecture Figure1(a) which corresponds to these physical properties of sequentially stained tissues. First, the model is optimized using optical density. Second, learning the virtual stain concentration map is separated from learning the virtual stain color, determined by the absorption coefficient vector.

The first step of our model involves transforming an input patch stained with stain 1 to the optical density domain, followed by a U-Net neural network (Ronneberger et al., 2015) for inferring a single-channel stain 2 concentration map for the patch. Subsequently, the concentration map is multiplied by a learnt absorption coefficient vector to derive an optical density virtual stain. For brightfield (BF) WSIs, the absorption coefficient is length three RGB vector, hence we call this model a 3x1 architecture. The optical density virtual stain is then added to the optical density input patch, which yields an optical density virtually stain-multiplexed patch. This patch is subsequently inverse-transformed to a virtually stain-multiplexed bright-field-like output. It is important to note that although the RGB vector is learnt during training, it is a global parameter vector and not dependent on the input patch.

The physical properties of sequential staining have also informed the development of our loss function for model optimization, as depicted in Figure. 1 (b). Given the additive nature of sequential/virtual staining, semantic masks, denoted as $Mask_{GT}/Mask_{out}$, are constructed for the ground-truth and output patches, respectively. These masks are obtained by taking the difference between the optical density ground-truth/output patch and

the input patch; any pixel with an optical density difference exceeding a threshold value is considered positive for sequential/virtual stain. The semantic masks are then used to compute a semantic pixel-wise binary cross entropy loss (BCE) between the corresponding ground-truth and output patches.

The semantic loss is combined with a mean squared error (MSE) loss between the optical density ground-truth and output patches. To focus the training on relevant differences, $Mask_{GT}$ is used as a mask for the MSE loss. The parameter $\alpha$ allows for tunable weighting of the relative strengths of these two loss functions, resulting in a combined loss function:

$$Loss = MSE(output, GT; Mask_{GT}) + \alpha BCE(Mask_{out}, Mask_{GT}) + \theta(-v)v^2 \quad (1)$$

Where the last term is a regularization term, reflecting the additive nature of sequential staining, keeping the inferred virtual stain output positive,$\theta$ is the Heaviside step-function and $v$ denotes the virtual stain output. We evaluate the model's performance using the intersection over union (IOU) metric between $Mask_{GT}$ and $Mask_{out}$, which is used for both hyperparameter tuning and computational evaluation.

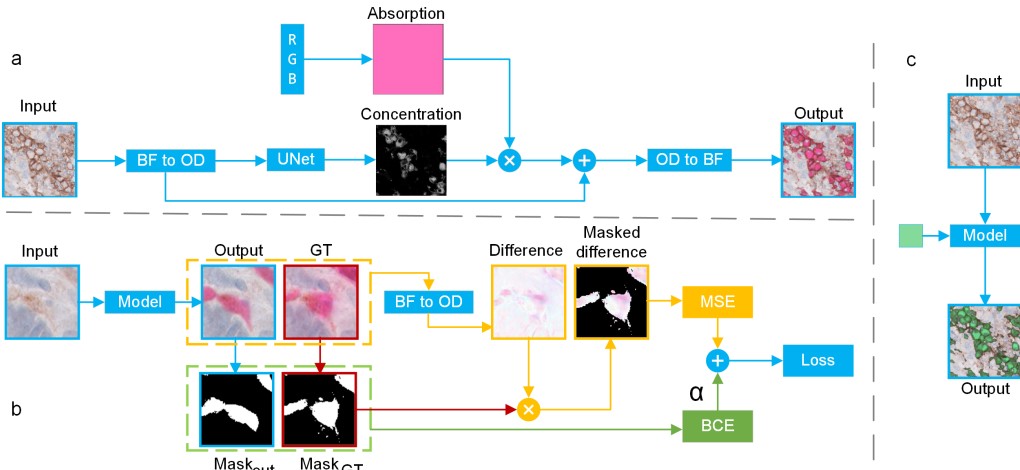

Figure 1: Model architecture and training loss structure. Stain 1 is PD-L1 22C3 pharmDx and stain 2 is CD68 **a.** 3x1 model architecture guided by physical staining process; virtual stain concentration inferred using a U-Net is combined with globally learnt virtual stain absorption coefficients vector in optical density (OD) domain, in accordance with Beer-Lambert law. **b.** Predicted virtually stained (output) and sequentially stained ground truth (GT) patches are used to compute a weighted combination of MSE and BCE losses. **c.** With the 3x1 architecture, the virtual stain hue can be easily changed at inference time.

## 3 Experimental Results

To demonstrate the utility of virtual stain multiplexing, we applied it to the problem of detecting macrophages in PD-L1 IHC 22C3 pharmDx stained NSCLC WSIs.

### 3.1 Dataset Preparation

Out of 200 NSCLC PD-L1 IHC 22C3 stained slides, 49 slides were chosen based on their TPS values, that were near to the clinically relevant thresholds of 1% and 50%. Following selection, the tissues were sequentially stained with CD68 PG-M1 (GA613), yielding sequentially stained paired WSIs. Each WSI pair was aligned; since paired WSIs are of the same tissue section, the obtained alignment was pixel-perfect. The study pathologist annotated tumor regions for each WSI pair which were used for model training. Of these, 26 regions where macrophages were difficult to distinguish from tumor cells, or where macrophages were identified as infiltrating PD-L1 positive tumor areas, were selected as validation-set regions for method evaluation and excluded from model training. See full dataset curation, preparation and annotation details in appendix A.

### 3.2 Training Details

We implemented the model described in section 2 using Pytorch. See full training and implementation details in appendix B.

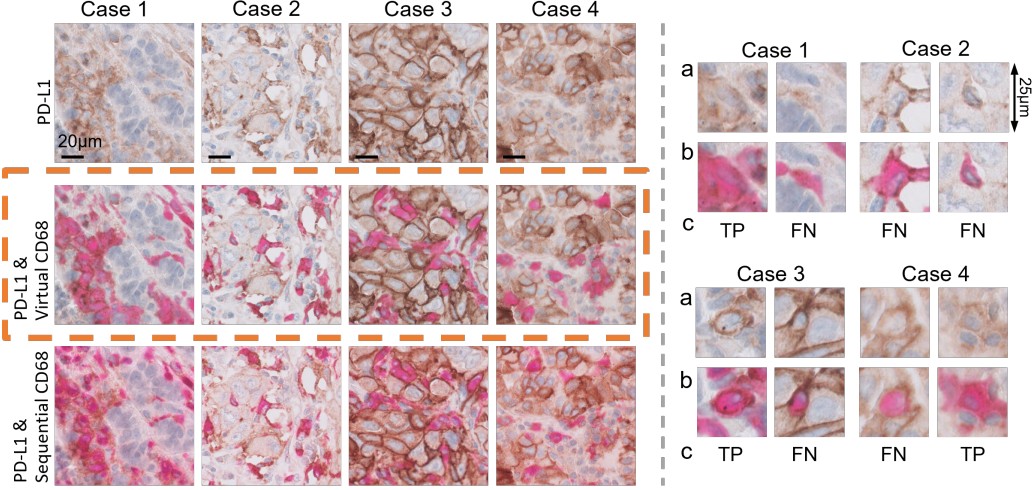

Figure 2: Left: comparison of virtual stain multiplexed and ground truth patches. Right: **a.** PD-L1+ cells annotated as CD68+ based on sequential CD68 staining by Pathologist 1 (ground truth). **b.** The virtual stain-multiplexed CD68 model correctly stains the tumor-infiltrating macrophages. **c.** Annotation of macrophages by Pathologist 2, based on PD-L1 staining only. The results are compared to ground-truth annotation and presented as True Positive (TP) and False Negative (FN).

### 3.3 Qualitative evaluation

To evaluate the performance of our proposed virtual stain-multiplexing method in whole-slide images (WSIs), we conducted a qualitative assessment. We present several representative examples of our model outputs generated on challenging validation set regions in Figure 2(left), alongside the corresponding input and sequentially stained ground truth patches. Notably, the virtual CD68 staining demonstrated a high degree of visual consistency with the actual CD68 stain, as evaluated by an expert pathologist. Additionally, our virtual stain method effectively highlighted macrophages, as evidenced by the staining of most macrophages in the validation set regions.

Figure 2(right) presents several examples of macrophages correctly identified by the model, despite the high variation in macrophage morphology and staining patterns. Not all these macrophages were correctly identified by pathologist 2 without the model's assistance.

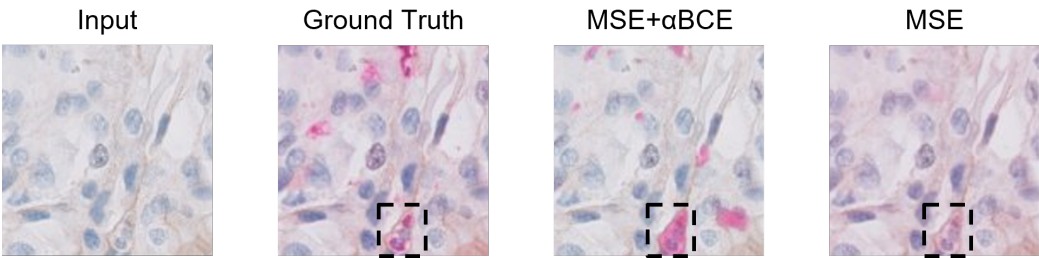

Figure 3: Qualitative effect of combined loss. Comparing (left to right) input, ground-truth, full model output and the output of an ablation model trained only using MSE loss. Dashed rectangles indicate a macrophage with clear membranal staining, which is partially stained in ground-truth patch, fully stained in full model outputs, and only faintly stained in the ablation model outputs.

### 3.4 Ablation Tests

**Qualitative effect of combined loss**
In Figure 3, we present an illustrative example that compares a ground truth patch and the corresponding output of a full model trained with combined MSE and BCE loss, to the output of an ablation model that is solely trained using MSE loss. The comparison shows that the outputs of the ablation model possess two distinctive features in contrast to those of the full model. Specifically, the ablation model acquires faint and non-specific background staining artifacts that are present in the ground-truth patches but are not present in input patches. Nevertheless, these non-specific stains do not appear in the outputs of the full model. Moreover, the outputs of the ablation model exhibit fainter overall virtual staining in comparison to the outputs of the full model and the ground truth patches. We observed that for cells with explicit membranal staining, the full model tends to stain the entire cell cytoplasm, while the ablation model often displays partial staining patterns, similar to those found in the ground truth patches.

**Quantitative ablation tests**

Several ablation tests were conducted to investigate the contributions of different parts of our proposed method. For full ablation tests details see appendix C.

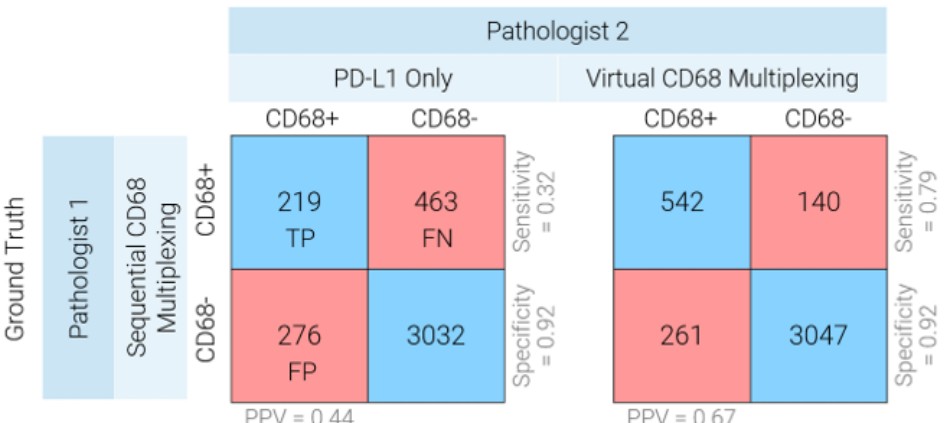

Figure 4: Cell classification model evaluation. Presented are confusion matrices comparing pathologist 2 annotations to pathologist 1 ground truth annotations, with (right) and without (left) the aid of virtual stain.

### 3.5 Cell-level evaluation

The aim of this evaluation was to directly test the potential of the suggested model as an assistive tool for pathologists in detecting macrophages. This evaluation scheme differs from a pixel-level evaluation, as it takes into account that the virtual stain may not always match the sequential ground truth stain pixel for pixel, but may still accurately stain the correct cells.

To carry out this evaluation, individual cells from 13 regions of interest, selected from the validation set, were annotated as macrophage/not-macrophage. Pathologist 1 generated cell-level ground-truth annotations based on matching pairs of PD-L1 & PD-L1 + sequentially stained CD68. Pathologist 2 first generated baseline annotations, based on PD-L1 WSI only. Then, pathologist 2 generated annotations based on matching pairs of PD-L1 & PD-L1 + virtually stain multiplexed CD68, viewed side-by-side. These annotations were then compared to the ground truth annotations by pathologist 1. The model performance was evaluated by measuring the change in pathologist 2's precision, sensitivity, and accuracy when assisted by the model com-pared to when not using the model.

Confusion matrices comparing the pathologist 2's annotations are presented in Figure 4. The addition of virtually stained multiplexed CD68 improved the performance of pathologist 2's annotations, increasing precision and sensitivity from 0.44 and 0.32 to 0.67 and 0.79, respectively, while specificity remained unchanged at 0.92. Paired McNemar's test (McNemar, 1947) yielded a p-value of less than $10^{-30}$.

## 4 Conclusions

In this pilot-study, we propose the method of virtual stain multiplexing, which combines virtual staining and virtual multiplexing. Virtual stain multiplexing of immunostains in the same section offers a promising avenue for the development of more accurate and reliable scoring methods for cancer diagnosis, without incurring additional reagents or tissue.

We presented a deep learning based model for CD68 virtual stain-multiplexing that can identify and virtually stain macrophages on an internal validation set. By virtually staining these cells, our model provides a useful tool for analyzing and interpreting PD-L1 22C3 pharmDx IHC WSI. Notably, we found that pathologist 2's performance in detecting macrophages improved significantly when assisted by the model.

In section 3.4, we demonstrated the importance of incorporating semantic loss in the ablation test. This addition was essential in enabling the model to focus on learning the relevant differences between the input and ground truth. The semantic BCE loss served as a guide for the model by providing information on where to add virtual staining, while the masked MSE loss directed the model on how to add it.

Interestingly, we observed that the addition of the semantic loss caused the model to virtually stain the entire cell cytoplasm, at the expense of visual fidelity to the ground truth staining patterns. However, this characteristic of the model was found to be advantageous in the context of macrophage detection. The study pathologists found it easier to interpret the virtual stain CD68 compared to the ground-truth sequential stain, owing to this behavior of the model.

In Section 3.5, we presented a quantitative pilot-study evaluation of the use of our model for macrophage detection. The results were encouraging, showing significant improvement in both precision and sensitivity of pathologist 2 when assisted by the model. In an ongoing study, building on this pilot-study, we will test the efficacy of the method with multiple pathologists.

Our proposed method can be extended to virtually multiplex additional stains, including those for other types of immune cells. As shown in Figure 1(c), the 3x1 architecture allows for arbitrary hue settings at inference time, facilitating such multiplexing. Moreover, our method opens up avenues for detailed exploration of the spatial arrangement of tumor infiltrating immune cells within the tumor microenvironment. Since our model utilizes the same input as the standard PD-L1 IHC 22C3 pharmDx stained tissue scans that are used in clinical practice, such investigations can be retrospective, employing clinically obtained data. This could lead to valuable insights into the relationship between immune cells and tumor cells, ultimately aiding in the development of new cancer therapies.

## Appendix A. Dataset Preparation

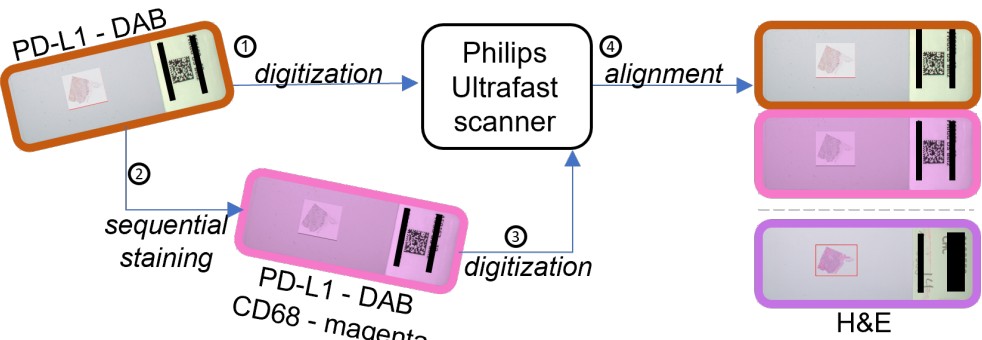

Figure 5: 200 NSCLC slides were stained with PD-L1 IHC 22C3 PharmDx (GE006). 49 selected tissues were sequentially stained with CD68 PG-M1 (GA613) and visualized with Envision FLEX HRP Magenta chromogen (GV925) on top of the PD-L1 IHC 22C3 pharmDx. An additional section was prepared for each case, stained with Hematoxylin and Eosin (H&E). All stained slides were scanned using a high-resolution scanner at 40x magnification. The sequentially stained whole-slide images (WSIs) were aligned with their matching WSIs to near pixel-perfect alignment. The H&E WSI were also aligned with their matching WSI pair using rough global alignment.

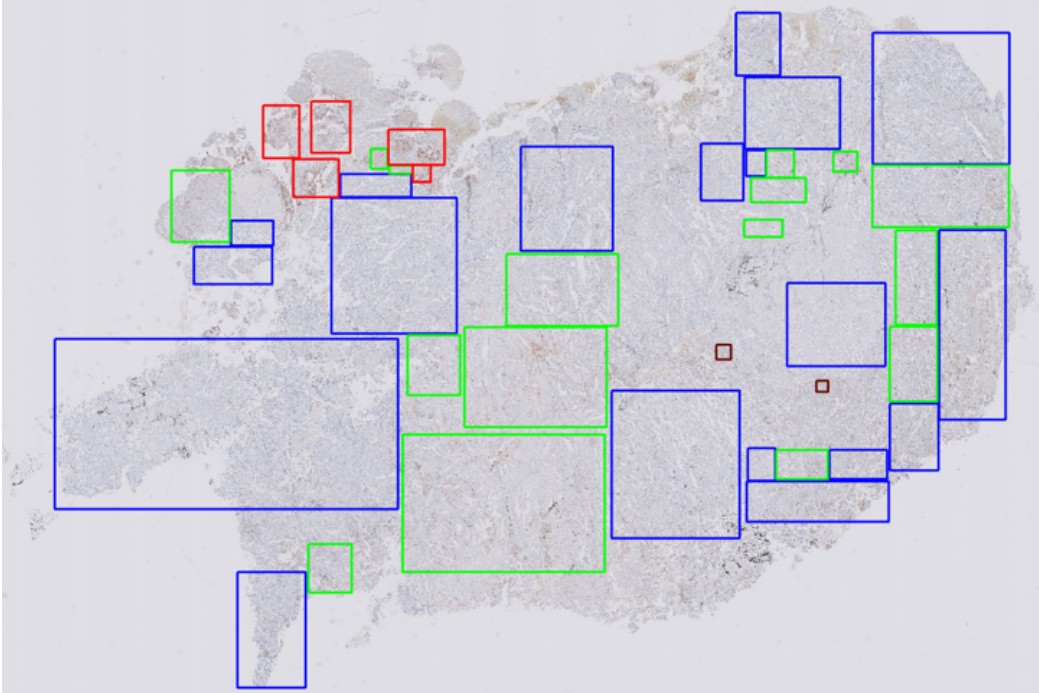

Figure 6: Tumor regions annotation: each case was annotated by the study pathologist, marking tumor and tumor-adjacent regions. Tumor identification was done by aligning the H&E and PD-L1 IHC 22C3 pharmDx stains. Regions were qualitatively classified according to common pathology practice: negative PD-L1 tumor (0 blue), weakly positive PD-L1 tumor (1+ green), and strongly positive PD-L1 tumor (2+, 3+ red). 1000x1000px regions from 26 WSIs, where macrophages were difficult to distinguish from tumor cells, or where macrophages were identified as infiltrating PD-L1 positive tumor areas, were annotated as validation-set regions for method evaluation and excluded from model training (brown)

## Appendix B. Architecture and Training Details

| Parameter | Value |
|---|---|
| **Pytorch version** | 1.13.1 |
| **Input patch size** | 512x512 pixels |
| **Concentration** network | U-Net |
| Depth | 7 |
| Min/Max channels | 64/1024 |
| Initialization | Xavier normal, with 0.01 gain |
| **Semantic loss weight** $\alpha$ | 0.1 |
| **Optimizer** | Adam |
| Learning Rate | 0.0002 |
| Betas | 0.9, 0.999 |
| **Scheduler** | ReduceLROnPlateau |
| Drop factor | 0.5 |
| Patience | 4 epochs |
| **Early stopping grace period** | 12 epochs |

Table 1: The model optimized using Adam optimizer and a reduce-on-plateau learning-rate scheduler, with early stopping. All hyperparameters were manually tuned; the hyperparameter choice used for cell-classification evaluation was guided by visual inspection of the produced virtual stain by pathologist 1 and validation set IoU. The additional initialization gain was required due to input transformation to optical density. Scheduler and early stopping were based on validation IoU metric. Data sampling was balanced using the region annotations, ensuring each batch included the same number of patches from each region class

## Appendix C. Quantitative Ablation Tests

| Architecture | Validation IoU mean (std) |
| --- | --- |
| Baseline | 0.617 (0.0015) |
| No semantic loss ($\alpha = 0$, unmasked MSE) | 0.631 (0.0015) |
| No OD transform | 0.608 (0.002) |
| No 3x1 architecture (simple U-Net) | 0.618 (0.0012) |
| No OD transform & No 3x1 architecture | 0.58 (0.0023) |

Table 2: Each test repeated five times, with different random seeds. Validation IoU values were averaged over last 10 epochs of each run, after convergence. To ensure stable evaluation, the learning schedule was fixed and early stopping disabled. The learning rate schedule was 0.0002 for 167 epochs and then linearly reduced by a factor of 100 over additional 84 epochs. Although removing semantic loss improved IoU, the visual qualities of the resultant stain were difficult for pathologists to interpret. Removing the OD transformation has a strong effect, which is amplified by replacing the 3x1 architecture with a simple U-Net.

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
