# OpenReview forum: "Deep-Learning Based Virtual Stain Multiplexing Immunohistochemistry Slides – a Pilot Study"
_MICCAI.org/2024/Workshop/COMPAYL — COMPAYL 2024_

### Official Review · Reviewer_B87b · 2024-07-03
**Virtual staining of mIHC using DL**

**Custom Rating:** 2
**Confidence:** 2

**Review:**

The authors introduce a DL-based method for virtually staining mIHC slides using a U-Net based model with a custom loss function. The model have been qualitatively evaluated with two pathologists.

Below are my comments:
1. Firstly, the paper needs a proper proofread to improve the clarity of prose and succinctness of sentences.
2. GE0006 should be cited when first mentioned.
3. Literature review can be significantly extended and improved as only one related work is mentioned.
4. The contributions of the work are not clearly outlined and how novel is the solution is not specified.
5. The motivation behind virtual staining should be more clearly described.
5. I think claiming that the architecture is novel and mentioning that multiple times in the paper is an over-statement. The model uses a U-Net and modified MSE loss function. I'm not sure how novel is that. Also, in the abstract, the novelty is exactly specified: "To this end, we designed a novel model architecture, guided by the physical sequential staining process which provides superior performance." Then the mention of a custom loss function comes next: "The model was optimised using a custom loss function that combines mean squared error (MSE) with semantic in-formation, allowing the model to focus on learning the relevant differences be-tween the input and ground truth." When I read the two former sentences, I perceive that the architecture is novel plus that the loss function is custom. But from reading the paper, it seems that this is not the case.
6. The conclusion section is too long and can be partially moved to a discussion section.
7. In the results section, there is a sparsity of numerical metrics through quantitive evaluation. I think this should be worrying.

---

### Official Review · Reviewer_26Kn · 2024-07-09
**Interesting approach to virtual staining, but the method is not properly validated**

**Custom Rating:** 3
**Confidence:** 4

**Review:**

Summary of paper:

The manuscript proposes a new method of virtual staining using a U-Net to predict an optical density map based on an input optical density map of another immunohistochemistry stain. In this work, the authors use PDL1 IHC as input stain and predict CD68 to detect macrophages and reduce the number of false positives for PDL1 counting.

Pros:
- The manuscript is well written, with great figures explaining the model and results.
- Their proposed solution including a mask for virtual staining is a nice way of creating cleaner output images.
- The authors include 2 pathologists to do a small practical evaluation demonstrating the usefulness of their approach.


Major concerns:
- The model is trained and validated on the same cohort, where staining and tissue preparation are similar. If the authors want to show that their approach really works, they must include an additional test set from a different source. Moreover, it is not clear if the validation regions are from WSI that were also used for training (in other regions).
- The validation areas are selected to be challenging regions. How does the model perform on less “challenging” regions where it should not predict false information?
- A threshold is mentioned to create the stain mask. How was this threshold determined? How does changing this threshold affect the model (during training and in general?) Does this threshold generalize?


Minor concerns:
- How much do the pathologists agree if they both use the real sequential staining?


Other comments:
- The manuscript does not follow the submission guidelines and is not using the correct template. Please use the provided template.


Code, model weights, and/or data availability:
- unavailable


Conclusion

The authors propose an interesting method for virtual staining and include a pathologist validation of their application which is always nice. However, the validation of their proposed approach is restricted to a single cohort on which the model was also trained on, and only on ROIs that were manually selected for this study.

---

### Official Review · Reviewer_2kVN · 2024-07-09
**Paired image translation between PD-L1 and CD68**

**Custom Rating:** 3
**Confidence:** 3

**Review:**

The paper introduces a method for stain translation from PD-L1 to CD68 to allow the visualization of macrophages on PD-L1 stained images and the removal of these towards a more accurate estimation of the Tumor Proportion Score. The transformation between PD-L1 to CD68 is modeled by a so-called 3x1 model architecture that separates the learning of the virtual stain concentration and the learning of the stain absorption coefficients. This is a alternative to other similar works relying on Pix2pix or CycleGAN. The method employs guidance through semantic segmentation through the minimization of a binary cross entropy loss on thresholded virtual / real images in addition to the use of a regression loss on the RGB channels between the virtual and ground truth PD-L1 / CD68 combined images.

- The paper is well written, the methods and the experiments clearly presented.
- The proposed method is showed to increase concordance between pathologists when classifying cells.
- It would be interesting to investigate whether this translates in an increase in the concordance of the TPS scores as well. Also, given the data at hand it would be interesting to develop and include an automatic detection of macrophages on PD-L1 stained images into an existing method for potentially improving on the automatic assessment of TPS / TC scores on these images.

- Given that paired images with near pixelwise accuracy are available, it is unclear why a pix2pix approach is not used to generate a in silico CD68 image from the co-registered PD-L1 image.
- The paper misses to cite previous works on virtual staining for PD-L1 scoring, in particular [Kapil et al., IEEE TMI 2021] and translation guidance based on semantic segmentation [Mahapatra et al, MICCAI 2020].

---

### Decision · Program_Chairs · 2024-07-16

Accept